# High-Value Oil–Water Separation Materials Prepared from Waste Polyethylene Terephthalate

**DOI:** 10.3390/molecules28227503

**Published:** 2023-11-09

**Authors:** Changjian Zhou, Jiahao Zhang, Yuqing Fu, Maowan Wu, Heng Zhang, Qingle Shi, Yong Dai, He Zhao

**Affiliations:** 1School of Chemistry and Chemical Engineering, Yancheng Institute of Technology, Yancheng 224051, China; zcj@ycit.cn (C.Z.);; 2Sunlour Pigment Co., Ltd., Xinghua 225431, China

**Keywords:** waste PET, recycle, water/oil separation, high-value reuse

## Abstract

As one of the most common forms of waste, waste PET is a serious pollutant in natural and human living environments. There is an urgent need to recycle PET. For this study, the complete degradation of PET was realized at a low temperature. A lipophilic hydrophobic membrane was formed on the surface of a stainless steel mesh (SSM) using a simple dip coating method, and an oil–water separation material was successfully prepared. After loading with degradation products, the surface roughness of SSM increased from 19.09 μm to 62.33 μm. The surface changed from hydrophilic to hydrophobic, and the water contact angle increased to 123°. The oil–water separation flux of the modified SSM was 9825 L/(m^2^·h), and the separation efficiency was 98.99%. The modified SSM had good reuse performance. This hydrophobic modification method can also be used to modify other porous substrates, such as activated carbon, filter paper, foam, and other materials. The porous substrate modified by the degradation product of waste PET was used to prepare oil–water separation materials, not only solving the problem of white pollution but also reducing the dependence on non-renewable resources in the conventional methods used for the preparation of oil–water separation materials. This study provides new raw materials and methods for the industrial production of oil–water separation materials, which have important application prospects.

## 1. Introduction

Polyethylene terephthalate (PET) is a linear polymer formed from the condensation of terephthalic acid and ethylene glycol that has the advantages of high cost performance, creep resistance, friction resistance, fatigue resistance, and other advantages [1]. The global consumption of polyester still maintains a growth rate of about 5% [2], and China has become the world’s largest consumer of plastics and plastic products, with the plastics industry being an important pillar of the national economy. Most PET packaging materials are disposable and discarded as garbage after one use [3,4,5]. The good thermal stability of PET makes it difficult to be degraded by microorganisms, resulting in serious white pollution problems [6]. Therefore, the recycling and utilization of PET polyester has become one of the most pressing issues to solve in environmental pollution and the recycling and utilization of polymer materials. Tawfik et al. utilized dibutyltin oxide as a catalyst at atmospheric pressure and 190 °C to facilitate the ammonolysis degradation of PET waste. The end result, bis bis(2-hydroxyethylene) terephthalamide, has numerous possible uses, including the potential to be used in coatings and adhesives, and it can be also used as a source of various polyurethanes [7]. Zhang et al. reported a one-pot, low-temperature catalytic method that directly converts different grades of PE to liquid alkyl aromatics and alkyl naphthalene at 280 °C without adding solvents or hydrogen molecules and using *γ*-alumina-supported platinum as a catalyst, demonstrating how waste polyolefin can be a viable feedstock for the generation of molecular hydrocarbon products [8]. Guo et al. used Perkalite F100 as a nano-catalyst to catalyze the depolymerization of PET and found that the main product obtained after PET depolymerization was high-purity double (2-hydroxyethyl) terephthalate (BHET), and the BHET monomers could be directly used as the starting material to further polymerize PET [9]. Bäckström et al. created a range of ethylene terephthalate esters from PET using fast catalytic-free ammoniac microwave assistance. Moreover, the ethyl terephthalate that is produced during the chemical recovery of PET can be utilized as a plasticizer for polylactic acid or as a reactant in the free radical thioene reaction, which is used to prepare plastic films. Ammonolysis, a chemical recycling process for PET, has been shown to be a practical and adaptable choice for producing a library of valuable compounds that could be used in material applications [10].

In general, in addition to being directly buried in landfills [11], PET waste can be recycled through direct combustion [12], physical recovery [13,14], enzymatic degradation [15], and chemical recovery [16]. Usually, the landfill of waste plastics will not only occupy a large amount of land, but the occupied land also cannot be restored for a long period of time as the decomposition of landfill plastics is very slow, and the substances produced will pollute the land and groundwater, endangering the surrounding environment [17]. Direct combustion is simple, easy, low-cost, and the most extensive recovery method at present, but PET will emit a lot of the greenhouse gas CO_2_, which is not in line with the current dual-reduction policy, and indirectly pollute the environment. Physical recycling refers to the separation, cleaning, crushing, and other forms of pretreatment of waste PET to remove labels, dust, and other pollutants before heating and melting the PET to create particles and processing the PET into new products. The recycled particles recovered by the physical method are not suitable for manufacturing high-grade plastic products, and the mechanical properties and thermal stability of recycled plastics are greatly reduced compared with raw materials; in addition, the application range of recycled plastics is narrow, so it is subject to certain restrictions. Enzymatic degradation has shortcomings, mainly related to its high cost and immature technology, and the performance of plastic products degraded by this method cannot fully meet various consumer needs. At present, the chemical recovery methods of PET include hydrolysis, glycol alcoholysis, methanol alcoholysis, and ammonia hydrolysis methods [18,19]. Among them, hydrolysis is a very widely used method in the context of recovering PET [20,21]. Using the hydrolysis of the ester group, PET can be hydrolyzed and depolymerized under alkaline, acidic, or neutral conditions to produce terephthalic acid (TPA) and ethylene glycol (EG). TPA and EG are the common raw materials for the production of primary PET, thus achieving the chemical recycling of PET [22,23,24]. However, at present, the disadvantages of chemical monomer recovery include the associated harsh reaction conditions, low product yield, and difficulty in separating and purifying of the product. Hence, there is an urgent need for alternative, efficient, and environmentally friendly solutions to achieve plastic recycling.

Oil/water pollution from industrial sources and spills has already emerged as one of the most difficult issues contaminating natural water bodies and exacerbating the scarcity of oil resources. Long-term research on the effective treatment of oil and water pollution has been conducted, despite its challenges. Due to oil/water emulsions’ unique properties (small droplet sizes, low density, and high stability), many more challenges arise during handling. Many researchers have created synthetic “oil-removing” filter materials by fusing low-energy surface coatings with rough surface structures [25]. Unfortunately, most materials are produced by covering porous substrates such as metal mesh [26], textiles [27], or polymer films [28], which have shortcomings such as low flux, difficult preparation processes, or a rapid decrease in permeability with a hydrophilic or lipophilic layer [29]. The micropores of the substrates allow them to effectively separate incompatible oil and water, but it is difficult to separate water–oil emulsions. Thus, for effective emulsion separation, it is imperative to develop suitable materials with outstanding cost-effectiveness and wetting stability.

In this study, waste PET was used as the raw material; after swelling with ethanol, a hydrolysis reaction was carried out with acetic acid at 80 °C for 3 h, and the degradation product was used as a hydrophobic modification agent to modify stainless steel mesh. The structure and oil–water separation performance of the modified stainless steel mesh were tested. At the same time, the hydrolysis reactions carried out for this study did not involve the use of a catalyst; this was not only to protect the environment and save costs but also to reduce the reaction conditions and reduce the energy consumption of the whole process. Using the degradation products as hydrophobic modifiers enhances the recovery value of PET.

## 2. Results and Discussion

### 2.1. Surface Morphology Analysis of Modified SSM

The effect of product modification on the SSM surface morphology was observed via SEM, and the experimental results are shown in Figure 1. The surface of the unmodified SSM is smooth, and the pore size is about 0.8 μm. After soaking in the PET degradation solution, the PET degradation product adhered to the SSM, making the surface rough, and the PET degradation product “grew” along the stainless steel mesh. These changes will alter the oil–water separation effect of the material.

EDX was used to test the element composition of the surface of the sample; the load and distribution of the PET degradation product on the surface of the SSM are shown in Figure 1. The EDS pattern of the SSM showed the existence of C, N, and O, originating from the presence of very small amounts of organic impurities on the surface of the SSM (Figure 2a); the contents of the C and N elements were 2.07% and 8.90%, respectively. In contrast, the peak signal of the C, N, and O elements in the modified SSM was significantly higher than that of the SSM (Figure 2b), which was caused by the adhesion of the PET degradation products on the surface of the SSM. Element mapping revealed the uniform distributions of C, N, and O. In addition, the color of C, N, and O in the modified SSM was darker than that of the SSM, and the content of the C and N elements on the modified SSM surface increased to 31.95% and 15.17%, respectively, which is consistent with the results obtained via EDS characterization, indicating that the PET degradation products successfully adhered to the surface of the modified SSM.

The surface roughness of the modified SSM was analyzed and measured via confocal laser scanning microscopy, and the influence of the loading of PET degradation products on the surface roughness of the SSM was studied. The results are shown in Figure 3. The surface roughness of the unmodified SSM was 19.09 um. After modification by the PET degradation products, the roughness of the SSM was significantly increased to 62.33 um, which was in accordance with the SEM results. These effects can be attributed to the deposition of the PET degradation products, which affect the oil–water separation ability of the material. 

### 2.2. Surface Wettability Analysis of Modified SSM

Wettability is an important index of oil–water separation materials, as it can affect their oil–water separation ability. Therefore, the water contact angle of the PET degradation products before and after SSM modification was measured, and the experimental results are shown in Figure 3. After pretreatment, the surface of the SSM was hydrophilic, and when the PET degradation products were loaded, the SSM became hydrophobic, and the water contact angle was 123° (Figure 4). This is because the PET degradation products contain a large number of hydrophobic groups, which are adsorbed onto the SSM, increasing the surface roughness of the SSM, thus increasing the water contact angle of the material.

### 2.3. Analysis of Oil–Water Separation Efficiency of Modified SSM

The experimental results of the modified materials’ oil–water separation performance are shown in Figure 5. The separation efficiency of the unmodified SSM was only 10.3%. Under the action of its own gravity, the separation efficiency of the modified SSM reached 98.99%, and the flux reached 9825 L/(m^2^·h), many times higher than that of pressure-driven commercial separation membranes [30]. These results are due to the increase in surface roughness after loading PET degradation products, resulting in an increase in the water contact angle, which in turn increases the SSM’s hydrophobicity, enhancing its demulsification ability and improving its oil–water separation efficiency. The SSM modified with the PET degradation products achieved a good separation effect for oil and water, and at the same time, waste PET was degraded and recycled to produce a high-value product.

In order to verify the reusability of the modified SSM, the oil–water separation cycle experiment was carried out on the modified SSM, and the experimental results are shown in Figure 6. After 10 cycles, the separation efficiency of the modified SSM for the emulsion remained between 95.7% and 99.6%, and the flux remained above 7000 L/(m^2^·h). These test results show that the modified SSM has good recycling performance.

## 3. Conclusions

In conclusion, PET was pretreated with ethanol for swelling, and then the waste plastic was completely degraded in acetic acid solution at 80 °C for 3 h. The hydrophobic modification of a porous substrate by the degradation products was used to prepare oil–water separation materials. After the product was loaded onto the SSM, the surface roughness and water contact angle were increased, the separation flux of the oil–water mixture reached 9825 L/(m^2^·h), and the separation efficiency reached 98.99%. In addition, the separation efficiency did not significantly decrease after 10 cycles of repeated use. This method not only solves the problem of thermosetting plastic waste but also provides new raw materials for the preparation of oil–water separation materials and minimizes the use of petrochemical resources.

## 4. Materials and Methods

### 4.1. Materials

Ethanol (C_2_H_5_OH, analytically pure) was purchased from Chengdu Colon Chemical Co., Ltd., Chengdu, China. Acetic acid (C_2_H_4_O_2_, analytically pure) was purchased from by Shanghai Titan Technology Co., Ltd., Shanghai, China. Stainless steel mesh (SSM) was purchased from Zilianzhong (Guangzhou) Stainless Steel Co., Ltd., Guangzhou, China. The PET waste was Coca-Cola waste bottles, sourced from the Coca-Cola Company, Nanjing, China. The deionized water (DI) was self-made.

### 4.2. PET Degradation

The PET waste was washed with detergent, hot water, and ethanol and then put it in a drying box to dry. After drying, it was cut into pieces and crushed into 2.5 × 2.5 mm pieces with a grinder.

Approximately 4 g PET particles were weighed and put into a 100 mL three-mouth flask, and 30 mL anhydrous ethanol was added; after that, the three-mouth flask was placed on a magnetic stirrer and swelled at 70 °C for 5 h. After the reaction was completed, a suction filter bottle was used for filtration. The weight of the swelling PET (S-PET) after swelling was 8.6 g, and the swelling rate was 115%.

The S-PET tablets and acetic acid were added to a 50 mL three-mouth flask and reacted in a magnetic stirrer. The reaction temperature was set to 80 °C, and the reaction time was 3 h. After the reaction was completed, the reaction mixture was transparent, and the reaction vessel was cooled to room temperature.

### 4.3. SSM Modification

SSM was put into a 4 mol/L aqueous solution of HNO_3_ to remove the surface oxide and then heated in a water bath at 60 °C for 4 h. Then, it was washed with anhydrous ethanol for 3 min, dried, and set aside.

The modified SSM was obtained by immersing the pre-treated SSM in a PET degradation solution with a concentration of 10%, subjected to ultrasonic treatment for 20 min, allowed to dry naturally, and then dried in a 60 °C oven for 24 h (Figure 7).

### 4.4. Preparation of Emulsion

Oil/water emulsions with different droplet sizes were prepared by mixing 100 mL of chloroform containing different dosages of Span 80 and 1.5 mL water for different durations using the ultrasonic method. Taking an emulsion with a droplet size of 3.5 µm as an example, 0.4 g span 80 was added into 100 mL chloroform, followed by 1.5 mL water. The mixture was ultrasonicated for 3 min, and no demulsification was observed after 8 h.

### 4.5. Oil/Water Separation

The modified SSM was placed on the filtration device as the filtration film. A 10 mL volume of the oil/water mixture (1:1) was added into the device. The separation process was carried out under gravity (Figure 8). The water content in the filtrate was measured using the Karl Fischer method. In order to clearly observe the separation process, the oil was dyed using Oil Red O, and the water was dyed using methylene blue. The absorption time was observed from the test video. The oil/water emulsion was separated according to the procedure above, except 15 mL of the oil/water emulsion was used instead of the oil/water immiscible mixture.

The water content in the oil was determined using GB/T 11146-2009 [31] “Determination of Water Content in Crude Oil Carl Fischer Coulomb drops”. The separation efficiency (Re) of the emulsion is as follows:Re = (W_o_ − W_1_)/W_o_ × 100%
where W_o_ is the water content in the emulsion before separation, and W_1_ is the water content in the oil after separation.

### 4.6. Characterization of Modified SSM

The surface morphology of the sample was observed and analyzed via SEM using a Hitachi su8020 with an acceleration voltage of 15 kV; the components of the sample surface were analyzed via X-ray energy dispersion spectrometry (EDX). Before observation, the sample was fixed on the sample table with a conductive adhesive and treated with gold spraying. The contact angle (CA) was measured uusing a JC2000D2H contact angle tester manufactured by Shanghai Zhongchen Technical Equipment Co., Ltd., Shanghai, China. The surface roughness of the sample was determined using a LEXT OLS4100 laser scanning confocal microscope from Shanghai Fulai Optical Technology Co., Ltd., Shanghai, China. The modified SSM was placed on the sample table at 25 °C, and the contact angles of water and chloroform were determined separately. The amounts of water and chloroform used were both 3 μL, and the average values of 3 different monitoring points on the surface of the sample were obtained.

## Figures and Tables

**Figure 1 molecules-28-07503-f001:**
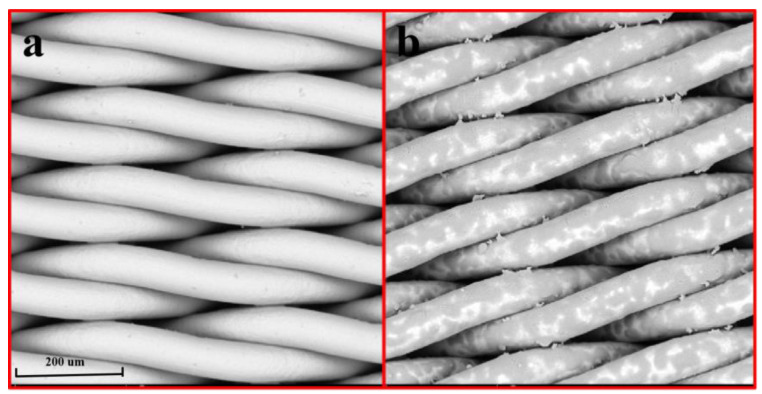
SEM images of (**a**) the SSM and (**b**) modified SSM.

**Figure 2 molecules-28-07503-f002:**
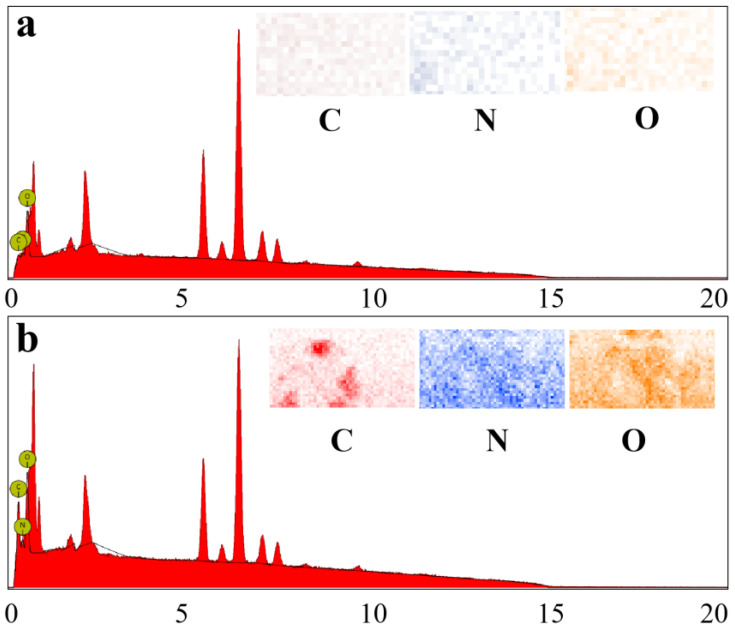
The EDX results and mapping of (**a**) the SSM and (**b**) modified SSM.

**Figure 3 molecules-28-07503-f003:**
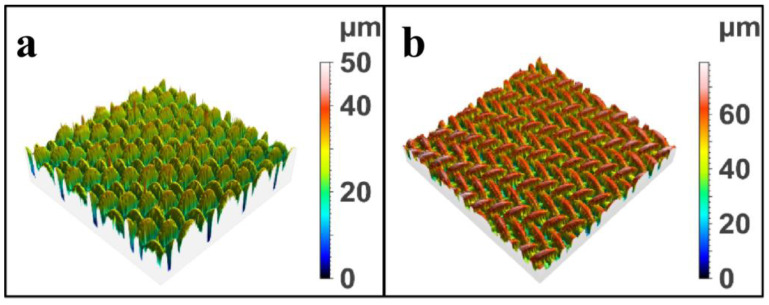
The laser confocal microscopy results of (**a**) the SSM and (**b**) modified SSM.

**Figure 4 molecules-28-07503-f004:**
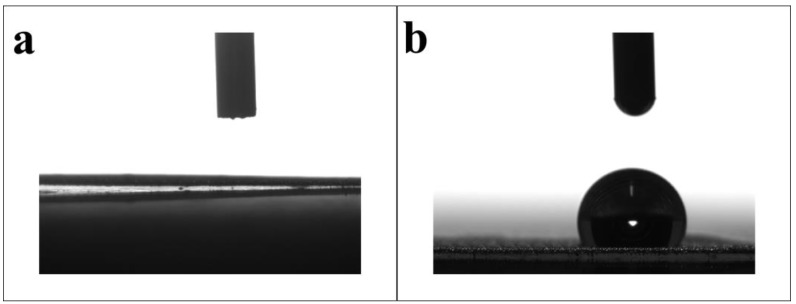
The water contact angles of (**a**) the SSM and (**b**) modified SSM.

**Figure 5 molecules-28-07503-f005:**
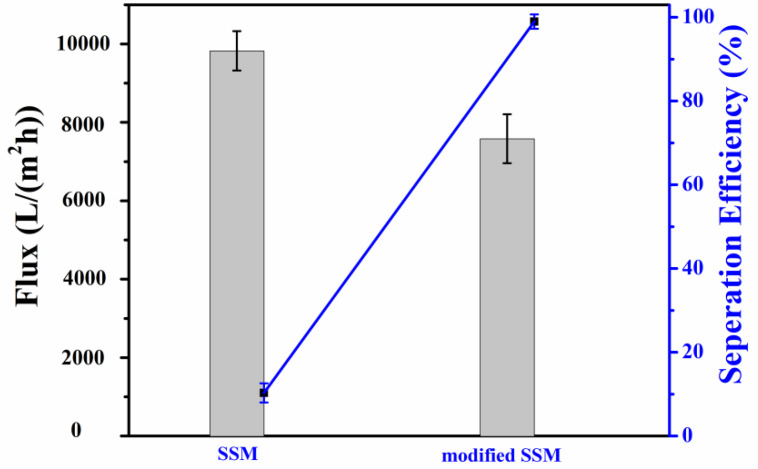
The flux and separation efficiency of the SSM and modified SSM for water and oil.

**Figure 6 molecules-28-07503-f006:**
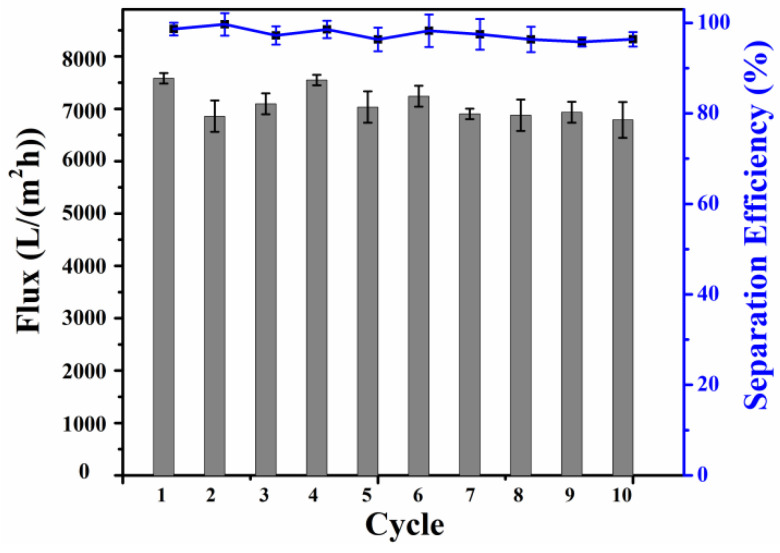
The flux and separation efficiency of the modified SSM during multi-cycle separation tests.

**Figure 7 molecules-28-07503-f007:**
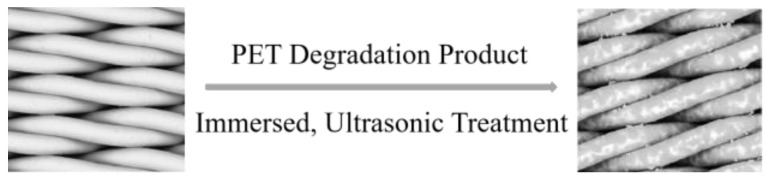
Preparation process for modified SSM.

**Figure 8 molecules-28-07503-f008:**
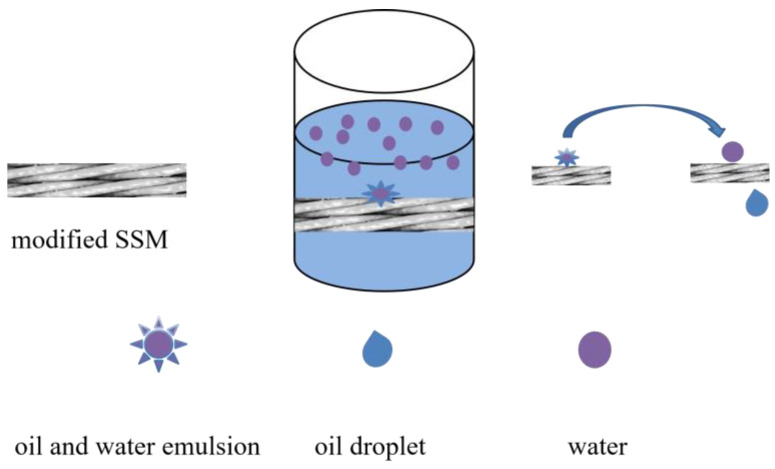
Schematic diagrams of the emulsion separated by the modified SSM.

## Data Availability

Data are contained within the article.

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
