# Peer review of "High-Value Oil–Water Separation Materials Prepared from Waste Polyethylene Terephthalate"

_molecules, 2023, doi:10.3390/molecules28227503_

Round 1

Reviewer 1 Report

Comments and Suggestions for Authors

The manuscript “High value oil-water separation materials were prepared from waste PET” is dedicated to the problem of PET waste. The level of novelty is high, but that’s where the advantages of the article end. The results are described briefly. There is no discussion of the data obtained and comparison with the works of other authors. The reference list is very short. The manuscript looks like a laboratory work, not an article in a Q1 journal.
In the discussion of the results, references to the work of other authors are always provided to compare the data and highlight the advantages of the work and results.

For a scientific article, reactions that occur during certain processes, process schemes, descriptions of kinetic and/or diffusion patterns, structural changes, and development of models of physical or chemical processes are important. For publication in a scientific journal, reaction equations, polymer destruction schemes, etc. should be added.

In the present form, the work is suitable for publication in a technological journal.

Author Response

Dear Reviewer:

Thank you for your valuable comments, your comments are quite helpful, and I revised my paper very carefully, thank you again for your help! If there is more question, we are willing to revise it again. We appreciate your warm work earnestly, and hope that the correction will meet with approval.

Given below is our specific point-by-point response to each of the specific comments.

Comments:

The manuscript “High value oil-water separation materials were prepared from waste PET” is dedicated to the problem of PET waste. The level of novelty is high, but that’s where the advantages of the article end. The results are described briefly. There is no discussion of the data obtained and comparison with the works of other authors. The reference list is very short. The manuscript looks like a laboratory work, not an article in a Q1 journal.

In the discussion of the results, references to the work of other authors are always provided to compare the data and highlight the advantages of the work and results.

For a scientific article, reactions that occur during certain processes, process schemes, descriptions of kinetic and/or diffusion patterns, structural changes, and development of models of physical or chemical processes are important. For publication in a scientific journal, reaction equations, polymer destruction schemes, etc. should be added.

In the present form, the work is suitable for publication in a technological journal.

 Response: Many thanks for your comments. According to your valuable advice, we have reworked the manuscript from beginning to end. The experimental results are supplemented and the data obtained are compared with those of other researchers.

In addition, we have supplemented the references covered in the background introduction. Please see line 40-102 and line 259-315.

In terms of text expression, we asked help from the language polishing function of MDPI, sorry for the bad experience brought to you.

In the article, according to your suggestions, we have added the relevant contents. Please see line 140 and 162 in our manuscript.

Once again, thank you very much for your comments and suggestions.

Sincerely yours,

Changjian Zhou

Reviewer 2 Report

Comments and Suggestions for Authors

This article reports on “High value oil-water separation materials were prepared from waste PET.” Below are comments that can  be addressed by authors to improve the manuscript quality.

Comments:

1.           Novelty (Line 32-39): The introduction establishes the significance and urgency of PET recycling due to environmental concerns. It would be beneficial if the authors could provide a clearer statement about the novelty of their research in the broader context of existing solutions.

2.           References (Line 31, 34, 38, 41): The authors have cited multiple references to support their statements. This adds credibility to their claims. It's essential to verify these references to ensure they provide appropriate context and support.

3.           Clarity (Line 40-43): The introduction mentions several methods of PET recycling. Elaborating on the advantages and disadvantages of each method, even briefly, would provide readers with a clearer understanding of where the authors' research fits in.

4.           Lack of Quantitative Data: The observations about surface morphology, while clear, seem qualitative. The section would benefit from quantitative measurements to support these observations. For instance, specifics on the degree of roughness, detailed pore size distribution, or statistical analyses would provide more robust results.

5.           Integration of Results with Analysis: While combining results with analysis can be efficient, it might make the section less clear for readers who are specifically interested in raw data. The authors might consider separating raw results from their interpretations for clarity.

6.           Dependence on Figures: The section seems to rely heavily on Figure 1. While figures are essential, the textual description should be comprehensive enough for readers to grasp the main findings even without the figure. Moreover, without viewing the figure, it's hard to assess its quality, but it's crucial that the figure be high-resolution, well-labeled, and accompanied by a clear legend.

7.           Contextual Relevance: The results presented discuss the change in surface morphology, but there isn't a clear link (in the provided excerpt) to why these changes are significant for oil-water separation. The broader implications of these findings in the context of the research problem should be emphasized.

8.           Comparative Analysis: It would be beneficial to compare the observed changes in surface morphology with other established methods or materials. Such a comparison would provide readers with a benchmark and highlight the novelty or superiority of the presented method.

9.           Reproducibility: The section would benefit from more detailed descriptions of the experimental conditions, ensuring that other researchers can reproduce the findings.

Comments on the Quality of English Language

It can be improved.

Author Response

26-Oct-2023

Manuscript ID: molecules-2683965

Title: High value oil-water separation materials were prepared from waste PET

Dear Reviewer:

Thank you for your valuable comments, your comments are quite helpful, and I revised my paper very carefully, thank you again for your help! If there is more question, we are willing to revise it again. We appreciate your warm work earnestly, and hope that the correction will meet with approval.

Given below is our specific point-by-point response to each of the specific comments.

Comments:

This article reports on “High value oil-water separation materials were prepared from waste PET.” Below are comments that can be addressed by authors to improve the manuscript quality.

  1. Novelty (Line 32-39): The introduction establishes the significance and urgency of PET recycling due to environmental concerns. It would be beneficial if the authors could provide a clearer statement about the novelty of their research in the broader context of existing solutions.

Response 1: Many thanks for your comments. We added relevant content in the background and summarized the innovation of our work at the end of the paragraph. Please see line 40-59, line 63-66, line 76-78 and line 88-102.

  1. References (Line 31, 34, 38, 41): The authors have cited multiple references to support their statements. This adds credibility to their claims. It's essential to verify these references to ensure they provide appropriate context and support.

Response 2: Thanks a lot for your constructive suggestions. We have verified the quote you said and adjusted the reference. Please see line 40-60.

  1. Clarity (Line 40-43): The introduction mentions several methods of PET recycling. Elaborating on the advantages and disadvantages of each method, even briefly, would provide readers with a clearer understanding of where the authors' research fits in.

Response 3: Many thanks for your comments. Based on your suggestions, we have elaborated on the advantages and disadvantages of each approach in order to give readers a clearer understanding of our research. Please see 63-78.

  1. Lack of Quantitative Data: The observations about surface morphology, while clear, seem qualitative. The section would benefit from quantitative measurements to support these observations. For instance, specifics on the degree of roughness, detailed pore size distribution, or statistical analyses would provide more robust results.

Response 4: Many thanks for your comments. We obtained quantitative data through roughness testing. Please see line 178-178, line 182-194, line 199-204, line 213-216 and line 224-227.

  1. Integration of Results with Analysis: While combining results with analysis can be efficient, it might make the section less clear for readers who are specifically interested in raw data. The authors might consider separating raw results from their interpretations for clarity.

Response 5: Many thanks for your comments. According to your suggestions, we have adjusted this part of the content.  

  1. Dependence on Figures: The section seems to rely heavily on Figure 1. While figures are essential, the textual description should be comprehensive enough for readers to grasp the main findings even without the figure. Moreover, without viewing the figure, it's hard to assess its quality, but it's crucial that the figure be high-resolution, well-labeled, and accompanied by a clear legend.

Response 6: Many thanks for your comments. we have enriched the text description and added the necessary instructions below the Figures so that readers can more easily understand the content of our articles. Please see line 178-178, line 182-194, line 199-204, line 213-216 and line 224-227.

  1. Contextual Relevance: The results presented discuss the change in surface morphology, but there isn't a clear link (in the provided excerpt) to why these changes are significant for oil-water separation. The broader implications of these findings in the context of the research problem should be emphasized.

Response 7: Many thanks for your comments. Loaded with the PET degradation products, the surface roughness of SSM increases, resulting in an increase in its water contact angle, which in turn makes its hydrophobicity stronger, demulsification ability enhanced, and oil-water separation efficiency improved. We have also added this description to the appropriate position in the manuscript.

  1. Comparative Analysis: It would be beneficial to compare the observed changes in surface morphology with other established methods or materials. Such a comparison would provide readers with a benchmark and highlight the novelty or superiority of the presented method.

Response 8: Many thanks for your comments. Compared with the commercial separation membrane, our material has a better oil-water separation effect. We have added this part in the text, please see line 200. Speaking of the innovation and advantage of this paper, our work lies in the waste plastic after treatment, its value increased, we are not only to prepare a kind of oil-water separation material, we start from a broader vision, more is to provide a method for the high-value application of waste PET.

  1. Reproducibility: The section would benefit from more detailed descriptions of the experimental conditions, ensuring that other researchers can reproduce the findings.

Response 9: Many thanks for your comments. According to your suggestion, we have added the relevant experimental conditions to the text so that other researchers can reproduce the findings. Please see line 139-157.

Once again, thank you very much for your comments and suggestions.

Sincerely yours,

Changjian Zhou

2023-10-30

Reviewer 3 Report

Comments and Suggestions for Authors

This work reported the stainless steel mesh coating by PET for oil-water separation. Such PET coated stainless steel mesh exhibited relative good chloroform-water separation efficiency. However, in my opinion, such work does not meet the standard for publishing in the journal of Molecules.

1.      In Experimentalsection, Lin 88, what does S-PET instead for? Line 71, 72, the molecular formulas should have subscripts. Line 109 contact Angle of water no need to capitalize A.

2.      How to measure the efficiency of oil-water efficiency? The experimental section as well as the main manuscript were not provided.

3.      The image and data qualities are poor. For example, due to the nonuniformity of the pore size of SSM, the average pore size need to be given according to Figure 1; The x axis of EDX data in Figure 2 can not be seen clearly. Incomplete box was leave in Figure 4.

4.      The concentration, soaking time need to be considered. Since the amount of degradative PET play important effects on both morphology of SSM as well as oil-water separation efficiency.

5.      Why does the author only choose chloroform to explore the oil-water separation capacity? How about other oils with different surface tension? More kinds of “oil” should be used in oil-water separation experiment to provide more comprehensive oil water efficiency.

Comments on the Quality of English Language

English language need to be improved.

Author Response

30-Oct-2023

Manuscript ID: molecules-2683965

Title: High value oil-water separation materials were prepared from waste PET

Dear Reviewer:

Thank you for your valuable comments, your comments are quite helpful, and I revised my paper very carefully, thank you again for your help! If there is more question, we are willing to revise it again. We appreciate your warm work earnestly, and hope that the correction will meet with approval.

Given below is our specific point-by-point response to each of the specific comments.

Comments:

This work reported the stainless steel mesh coating by PET for oil-water separation. Such PET coated stainless steel mesh exhibited relative good chloroform-water separation efficiency. However, in my opinion, such work does not meet the standard for publishing in the journal of Molecules.

  1. In “Experimental"section, Lin 88, what does S-PET instead for? Line 71, 72, the molecularformulas should have subscripts. Line 109 contact Angle of water no need to capitalize A.

Response 1: Many thanks for your comments. According to your prompt, we have made corresponding changes in the corresponding position of the text. Please see line 126, 114, 115 and 163.

  1. How to measure the efficiency of oil-water efficiency? The experimental section as well as themain manuscript were not provided.

Response 2: Thanks a lot for your constructive suggestions. According to your suggestion, we have added this part of the text in the corresponding position. The water content in oil was determined using GB/T 11146-2009 "Determination of Water Content in Crude Oil Carl Fischer Coulomb drops”. The water content in the emulsion before separation is Wo, the water content in the oil after separation is W1, and the separation efficiency (Re) of the emulsion is as follows: Re = (Wo-W1)/Wo * 100%. And we have added this part in our manuscript, please see line 153-157.

  1. The image and data qualities are poor. For example, due to the nonuniformity of the pore sizeof SSM, the average pore size need to be given according to Figure 1; The x axis of EDX data inFigure 2 can not be seen clearly. Incomplete box was leave in Figure 4.

Response 3: Many thanks for your comments. According to your suggestions, we have refined the images and data. After the modification of SSM, it can be seen from SEM characterization that the pore diameter of the modified material is reduced, which is caused by the coating of PET products on SSM. The x axis of EDX data in Figure 2 has been modified. The incomplete box left in Figure 4 has also been fixed. Please see them in line 179, 194 and 216.

Figure 1. The SEM of (a) SSM and (b) modified SSM.

Figure 2. The EDX and mapping of (a) SSM and (b) modified SSM.

Figure 4. The water contact angles of (a) SSM and (b) modified SSM.

  1. 4. The concentration, soaking time need to be considered. Since the amount of degradative PETplay important effects on both morphology of SSM as well as oil-water separation efficiency.

Response 4: Many thanks for your comments. In the pre-experiment, we roughly measured the efficiency of the material with a concentration between 1% and 20% for the separation of oil and water, and found that the material with a concentration of 10% had the best separation effect. So we use the 10% concentration directly in this paper. For the soaking time, we really did not consider the details, and directly treated the material with ultrasound for 20 minutes.

  1. Why does the author only choose chloroform to explore the oil-water separation capacity? Howabout other oils with different surface tension? More kinds of“oil" should be used in oil-waterseparation experiment to provide more comprehensive oil water efficiency.

Response 5: Many thanks for your comments. Chloroform is a representative of such a substance to determine the separation efficiency of oil and water, and many researchers have chosen to use this to determine the separation efficiency of oil and water. In this paper, we did not consider the influence of other oils on the surface tension of the material, but we will consider the influence of other oils on the surface tension of the material in the subsequent work.

Comments on the Quality of English Language

English language need to be improved.

Response: We have asked for the help of the language polishing function of MDPI.

Once again, thank you very much for your comments and suggestions.

Sincerely yours,

Changjian Zhou

2023-10-31

Round 2

Reviewer 1 Report

Comments and Suggestions for Authors

Accept in present form

Author Response

Dear reviewer:

Thank you very much for your help and comments on our articles. We will continue to examine our manuscript carefully. Thank you again.

Reviewer 2 Report

Comments and Suggestions for Authors

All comments have been properly addressed.

Comments on the Quality of English Language

It will benefit from additional proofreading.

Author Response

(The authors gave the same response as above.)

Reviewer 3 Report

Comments and Suggestions for Authors

This is my second time to review this manuscript. The authors revised the original manuscript under the reviewer's suggestion. However, in general, for one thing, such work lacks of innovation, since lots of oil-water separation materials based on both PET and stainless steel meshes have been reported. For anthor thing, the characterizations and researches are not systematic and comprehensive enough, e.g, the concentration, soaking time need to be considered; more kinds of “oil” should be used in oil-water separation experiment to provide more comprehensive oil water efficiency. What's more, the mechanism of  oil -water separation had not been clearly provided. In my opinion, this manuscript is not proper for published in Molecule in current state.

Comments on the Quality of English Language

English language can be improved.

Author Response

Dear Reviewer:

Thank you for your valuable comments, your comments are quite helpful, and I revised my paper very carefully, thank you again for your help! If there is more question, we are willing to revise it again. We appreciate your warm work earnestly, and hope that the correction will meet with approval.

Given below is our specific point-by-point response to each of the specific comments.

  1. This is my second time to review this manuscript. The authors revised the original manuscript under the reviewer's suggestion. However, in general, for one thing, such work lacks of innovation, since lots of oil-water separation materials based on both PET and stainless steel meshes have been reported.

Response 1: Many thanks for your comments. Speaking of the innovation and advantage of this paper, our work lies in the waste plastic after treatment, its value increased, we are not only to prepare a kind of oil-water separation material, we start from a broader vision, more is to provide a method for the high-value application of waste PET.

  1. For anthor thing, the characterizations and researches are not systematic and comprehensive enough, e.g, the concentration, soaking time need to be considered; more kinds of “oil” should be used in oil-water separation experiment to provide more comprehensive oil water efficiency.

Response 2: Thanks a lot for your constructive suggestions. From our research content, we will waste plastics for high-value utilization, oil-water separation is an application of this kind of material, and for this can be used for oil-water separation of the material itself, for its characterization is actually more adequate. The concentration, a relatively key influencing factor, has been explained in our manuscript. As for soaking time, it is not the key influencing factor, so it is not focused on the study. There is also the need for systematic research on the types of oil you said, for this point, we need to explain that the focus of our article is to use waste plastics to high value to transform, oil-water separation is an application of this material, our focus is indeed not on oil-water separation. I am very sorry for the bad reading experience. But your comments do play a very important role in improving the quality of our manuscript.

  1. What's more, the mechanism of oil -water separation had not been clearly provided. In my opinion, this manuscript is not proper for published in Molecule in current state.

Response 3: Many thanks for your comments. For this issue, we have to go back to the first and second questions you mentioned. Starting from our research content, we aim to make waste plastics capable of high-value utilization, and oil-water separation is only an application of this material. The focus of this paper is also to convert waste plastics to high-value utilization. To provide a potential path for the possible future use of waste plastics, our focus is really not on the oil-water separation aspect, so we did not conduct in-depth research on the mechanism of oil-water separation, which is really not our area of expertise. But if you must insist that we explain the mechanism, we are willing to try to learn. We hope you can give us a chance to explain.

English language can be improved.

Response: We have asked for the help of the language polishing function of MDPI to help us to improve the quality of our manuscript.

Once again, thank you very much for your comments and suggestions.

Sincerely yours,

Changjian Zhou

2023-11-5